# An Inquiry into the Relationship between Drug Users’ Psychological Situations and Their Drug-Taking Behaviour

**DOI:** 10.3390/ijerph182312730

**Published:** 2021-12-02

**Authors:** Gabriel Kwun Wa Lee, Gloria Chan, T. Wing Lo, Jerf W. K. Yeung, Cherry H. L. Tam, Xin Guan

**Affiliations:** Department of Social and Behavioural Sciences, City University of Hong Kong, Hong Kong; gabriel.lkw@cityu.edu.hk (G.K.W.L.); glo.c@cityu.edu.hk (G.C.); jerf.yeung@cityu.edu.hk (J.W.K.Y.); ss.hltam@cityu.edu.hk (C.H.L.T.); guan.xin@my.cityu.edu.hk (X.G.)

**Keywords:** drug use, soulmate, significant others, family, drug treatment, therapy

## Abstract

In view of the research gap whereby few studies have investigated the inner psychological situations underlying continuous drug use, this study used the Soulmate Scale to investigate the relationship between soulmate experience and drug-taking behaviour. Overall, 276 participants took part in this study. Results showed that soulmate experience was negatively related to drug-taking behaviour, which means that being psychologically attached to drugs and receiving comfort from them encourages dependency and a higher level of difficulty in quitting drugs. In addition, soulmate experience significantly mediated the effect of meaning of life and social isolation on drugs, suggesting that when such psychological bonding and sustenance can be developed in interpersonal relationships instead of drugs, drug users are likely to develop the meaning of life and a lower sense of social isolation, and are more likely to quit drugs. The corresponding implications were discussed.

## 1. Introduction

Reasons for youths’ drug-taking have often been attributed to their need to have fun, pleasure, and relieve boredom [1,2,3]. Despite this, these reasons cannot adequately account for the continuous use of drugs. According to Ahad and colleagues’ study on youth participants recruited from drug rehabilitation centres, the two major problems experienced by the participants included ‘being neglected by relatives’ (88.09%) and ‘being neglected by their family members’ (73.81%) [4]. This kind of neglect gave them a sense of social alienation, which instigated them to take drugs continuously [4]. Similarly, in Vanrozama and Gobalakrishnan’s [5] study on youth cases from de-addiction homes, participants experienced difficulty in quitting drugs and relapsed due to: (1) influence from the drug-taking peers; (2) the wish ‘to be part of the in-group’ (p. 1071); and also (3) the sceptical attitude, distrust, and rejection by significant others (e.g., spouses and family members) and the neighbourhood, in spite of their willingness to adapt a good, healthy life. These results show that it is their need for support and acceptance by other people and the frustration and loneliness experienced as a result of social isolation that underlies continuous drug use. 

A variety of therapies have been employed as an intervention for drug use, including Cognitive Behavioural Therapy (CBT), Motivation Enhancement Therapy (MET), and family therapy, which have been used for drug treatment (e.g., [6,7,8,9,10]). However, a major focus of these therapies is to change drug-taking behaviours rather than dealing with the causes that underpin the meaning of drug-taking. For example, CBT is a ‘problem-oriented’ type of therapy focussing ‘on the present’ and the improvement of the client’s ‘current state of mind’, rather than addressing ‘the causes of distress or symptoms in the past’ [11] (p. 580). Without treating the underlying causes of drug use, a probable result is repeated relapses. Hence, it is important to adopt an alternative perspective of looking at the problem of drug-taking, and to develop alternative methods to deal with this problem correspondingly.

This study seeks to enrich the research on drug use, adopting a soulmate perspective by placing a specific focus on the psychological situations of drug use. In this light, this study used the Soulmate Scale [12] to investigate the relationship between soulmate experience and drug use.

## 2. Literature Review

### 2.1. Reasons for Continuous Drug Use: A Soulmate Perspective

Existing research has pointed out various reasons for drug-taking, such as having a low level of self-control and resilience, relaxation, and the alleviation of negative emotions [13,14]. However, relatively little research has investigated the underpinning meaning of drug-taking from the drug users’ perspective. Chan and colleagues [15] adopted a *soulmate* perspective to investigate the reasons driving drug users to take drugs in a continuous manner. Based on Self-Determination Theory (SDT), the positive growth and development of an individual require the fulfilment of humans’ inborn psychological needs (i.e., ‘autonomy’, ‘competence’, and ‘relatedness’) and innate tendencies [16] (p. 68). Autonomy means the need to feel oneself ‘as the originator of’ one’s own decisions [17] (p. 3), while competence is referred to as ‘the need to feel capable of achieving desired outcomes’ [17] (p. 3), which is similar to self-efficacy [18]. Relatedness means ‘the need to be connected with others, accepted by others, to love and provide care for others as well as to be loved and cared for by others’ [15] (p. 2) [19,20]. 

Autonomy, competence, and relatedness can be understood with the following depictions. Regarding autonomy, it refers to ‘a means of determining and exercising authority over what it is that we care about in terms of our survival and our concerns for our future self’ [21] (p. 384). It is related to personal identity, featured by being distinguishable from other people and being ‘autonomous decision-makers with an agency power over their existence and becoming’ [22] (p. 2). Failing to achieve autonomy and identity, including the establishment of individual values, the ability to regulate own behaviour, and the ability to make independent decisions without being affected by other people, will increase one’s likelihood of taking drugs [23,24]. For competence, having a meaning in life, which is ‘an individual’s belief that he is fulfilling his positively valued life-framework or life-goal’ [25] (p. 409), could be regarded as a related concept. Having a life experience of ‘significance’, ‘coherence’, and being ‘worth living’ [26] (p. 7) is related to one’s likelihood of drug-taking. If one lacks a sense of meaning or purpose in life, s/he will take drugs to relieve the negative emotions [27]. 

For relatedness, Chan and colleagues [18] found that it is a crucial element determining one’s tendency to take drugs, which is separate from competence and autonomy. According to Baumeister and Leary [19], ‘the need to belong’ is a ‘fundamental motivation’ which people strive to fulfil: ‘People should show tendencies to seek out interpersonal contacts and cultivate possible relationships, at least until they have reached a minimum level of social contact and relatedness’ (p. 500). To fulfil such a need, ‘the person must believe that the other cares about his or her welfare and likes (or loves) him or her’; in this sense, the relationships formed with others are more than ‘mere affiliation’, but in addition, privilege mutual feelings [19] (p. 500). In addition, relationships, by nature, can substitute those which are lost in order to ‘overcome potential ill effects of social deprivation’ [19] (p. 500). If the need to belong or the need for relatedness is not satisfied, negative emotions, such as anxiety, depression, and loneliness, will likely arise [19]. These concepts apply to the context of drug-taking [18]. According to Chan and colleagues’ [18] study on 103 drug users recruited from drug treatment centres, in which those aged 21–30 were the majority, it was found that apart from one’s determination to quit drugs (i.e., autonomy) and having life goals (e.g., getting a job) which enhance a sense of self-worth and efficacy (i.e., competence), relationships (i.e., relatedness) was a core reason affecting the participants to engage in drug-taking. Participants took drugs because: (1) they failed to receive care and warmth from significant others and close companions, and even suffered from the loss of these people in their lives; and/or (2) it became a way to affiliate with other people, and maintain relationships. The negative emotions (e.g., sadness, despair) stimulated by the poor relationships with significant others, and even the loss of important ones in life (e.g., with spouses) drove them to take drugs to ‘hypnotize themselves’, to alleviate the trauma, and the feeling of being ‘collapsed’ [18] (p. 8). In this sense, drugs served as ‘psychological substitutes for the lack of connectedness in their everyday life’ [18] (p. 8). 

Psychologically, such a dependence on drugs resembles an affiliation with a soulmate [12]. Soulmate experience is characterised by a feeling of comfort, security, and fulfilment, being ‘at home’, and ‘an easily sustained deep intimacy with a lack of emotional barriers’ [28]. In this sense, taking drugs to achieve psychological sustenance could be compared to the experience of ‘seeking a soulmate for receiving comfort, a sense of security and satisfaction to relieve feelings of loneliness’, obtaining ‘a sense of belonging or “being loved”’ [12] (p. 2). Lo et al. [12] described the soulmate experience in drug-taking as ‘a close and emotional’ ‘spiritual attachment and connection between substances and users’, which is ‘is quite similar to falling in love with another person’, that is ‘always available and supportive in times of trouble and willing to listen to ventilation of concerns without any judgement or blame’ (p. 3). Drugs, as soulmates, give drug users ‘some sort of unconditional positive regard’, helping them ‘obtain relief and feel comfort and satisfaction’ in times of loneliness and when suffering from poor relationships with significant others [12] (p. 3). 

### 2.2. Effective Ways for Quitting Drugs 

Chan and colleagues’ study showed that significant others were ‘a double-edged sword’ (p. 10): relationships with significant others affected participants’ drug-taking behaviour in both negative and positive ways. Some participants had a strong desire to quit drugs because they received unconditional love, care, support, and trust from their significant others [18]. Such love, care, support, and acceptance brought them ‘hope’ and the ‘the motivation to live’ [18] (p. 10). This finding suggests that it is the fulfilment of the innate psychological need for ‘relatedness’ [15] (p. 68) that matters in the context of drug-taking. Having meaningful connectedness with others, as well as warmth and affection, helps fulfil the psychological need, stimulates positive emotions, and decreases the likelihood of drug relapse [18]. If such need has not been fulfilled, users will turn their needs for affiliation and connectedness into a craving for drugs; drugs become their ‘soulmates’, and ‘their spiritually trusted partners’ to form an attachment to, in order to achieve spiritual fulfilment [12] (p. 12). 

Summarising the above, social isolation has a negative impact on quitting drugs and likely triggers relapse, while having close relationships with significant others characterised by love, care, and support helps one quit drugs. In view of the importance of relatedness, it was suggested that drug treatments should nurture the psychological needs of the drug users by paying particular attention to the cultivation of their meaningful connections with significant others [18]. 

### 2.3. Current Therapies for Drug Treatment and Their Effectiveness

CBT, MET, and family therapy have been widely used for drug treatment (e.g., [6,10]). CBT is a problem-focussed kind of therapy that aims to help clients gain insights into their cognitive and behavioural patterns, as well as helping them develop ways to alter their maladaptive thinking and behaviours [11]. The therapy comprises the use of cognitive (e.g., ‘thought records’ which are used for keeping track of thought patterns and help to adopt the use of alternative ways of thinking) and behavioural (e.g., task assignments, relaxation) techniques [11] (p. 581). Applied as drug treatment, CBT can be provided as either individual or group intervention, with the use of relapse prevention techniques, such as the identification of contexts that instigate drug use (e.g., the presence of drug-taking peers), and the enhancement of capabilities to cope with the pull factors of drug use (e.g., a thorough understanding of the effects of drug-taking) [29]. For MET, there is a therapeutic approach which posits that clients will make most effective changes when they are intrinsically motivated instead of when they are imposed during therapeutic sessions [30]. Motivational interviewing (MI), developed by Miller and Rollnick (1991), is a key element of MET, which is ‘a person-centred goal-orientated approach for facilitating change through exploring and resolving ambivalence’ [31] (p. 138). MI operates on five principles: (1) express empathy (i.e., understand the client’s experiences, concerns, and difficulties faced); (2) develop discrepancy (i.e., help the client gain insights into ‘where they are and where they want to be’); (3) avoid argumentation (i.e., the therapist makes room for the client to freely express their feelings and ideas); (4) roll with resistance (i.e., the therapist adopts a ‘collaborative approach’ in which they understand the client’s perspective and ‘flow with the resistance’); and (5) support self-efficacy (e.g., the therapist acknowledges the client’s potential and efforts exerted during the course of making changes) [32,33]. Family therapy, which is based on the premise that family factors (e.g., parenting issues, family conflicts) are predictive of drug use [34], is provided to change the problematic patterns of the family system, as well as to make use of the support of the family to help the individual reduce drug-taking and achieve recovery [35]. Examples of family therapy include Behavioural Family Therapy, which emphasises ‘the role of the family in reinforcing behaviours and attitudes conducive to drug abuse, and attempt[s] to alter these contingencies so that the family can help promote the individual’s abstinence’ [36] (pp. 60–61). Functional Family Therapy (FFT) adopts a system perspective aimed at correcting maladaptive family interactions and encouraging positive family patterns through employing behavioural techniques so as to deal with youths’ problems [37]. Multidimensional Family Therapy (MDFT) posits that youths’ drug use is affected by multiple factors including the individual, family, peers, and community; it thus focuses on the functioning of the above aspects in terms of the youth, the family relationships, and the interaction between the family and various social systems (e.g., schools, welfare systems) [10]. 

Regarding the effectiveness of the above therapies for drug treatment on youth, Hendriks and colleagues (2011) found that both CBT and MDFT helped decrease adolescents’ problematic cannabis use. FFT was also stated as being effective in reducing youths’ drug use [38,39]. For MET or MI, Li and colleagues [8], adopting a systematic review and meta-analysis, noted that such therapy ‘was effective only in attitude change, not behavior change’ (p. 8). Regarding the comparison between therapies, Liddle [9] found that in adolescents who had received MDFT, their reduction in drug use continued during the 12 months after discharge, while for those receiving CBT, their reduction in drug use levelled off. Similarly, Liddle and colleagues [40] found that while both CBT (individual) and MDFT helped reduce cannabis use among youths, MDFT showed a more sustainable treatment effect than CBT in terms of the severity of the drug use problem. These findings suggest that drug treatment, which involves the family—particularly the improvement of relatedness with family members—is more able to deal with youths’ drug use problems. Helping drug users have good relationships with adaptive-functioning families serves as a buffer to relapse [41].

### 2.4. Present Study

Based on the framework of SDT, autonomy, competence, and relatedness are three central elements affecting one’s tendency to pursue healthy development and the engagement in drug-taking. Alongside the finding of soulmate experience (a kind of relatedness with drugs under the framework of SDT) on continuous drug-taking [12], this raises doubts as to whether soulmate experience with regard to relatedness, rather than autonomy and competence, is the ultimate explanation and authentic reason for persistent substance use.

Existing research has shown the importance of relatedness, particularly with significant others (e.g., family), in helping one reduce drug use (e.g., [15,41]). Relatedness helps reduce the sense of loneliness and the user to achieve a sense of well-being and fulfilment [15]. Particularly for youth, establishing affective connections with other people ‘is fundamental in developing the adolescent’s identity’ [42] (p. 3). Owing to the importance of establishing affective bonds as a fundamental psychological need of people in reducing drug use, it is important to investigate the psychological experiences of drug users. Despite the existence of research on drug use in the light of soulmate experience, such research is still scarce [12]. In order to fill this research gap, this study seeks to investigate the psychological experiences of drug users which affect their drug use. The psychological experiences of drug users are investigated quantitatively in terms of various factors, including (1) sense of identity; (2) social isolation; (3) soulmate experience; (4) meaning of life; and (5) self-efficacy, which encompass the aspects of autonomy, relatedness, and competence in the SDT framework (see Figure 1). 

Based on the above research objective, four hypotheses were developed as follows:The effect of a sense of identity on quitting drugs is mediated by the degree of soulmate experience.The effect of social isolation on quitting drugs is mediated by the degree of soulmate experience.The effect of meaning of life on quitting drugs is mediated by the degree of soulmate experience.The effect of self-efficacy on quitting drugs is mediated by the degree of soulmate experience.

## 3. Methodology

### 3.1. Sampling

A total of 276 drug users aged below 40 years old were recruited through four correctional institutions (2 male and 2 female) run by the Hong Kong Correctional Services Department. The institutions are used to detain inmates under the Drug Addiction Treatment Centres Ordinance, followed by aftercare supervision for 12 months upon release [43] (see Table 1). If they relapse during the supervision period, they will be recalled to the institutions for further treatment. In total, 160 participants came from CI-1 (95 inmates, 75 recallees, and 20 supervisees), 65 from CI-2 (34 inmates, 20 recallees, and 11 supervisees), 17 from CI-3 (13 inmates and 5 recallees), and 3 inmates from CI-4. Both CI-3 and CI-4 are youth institutions, so the samples are small.

All respondents were serving drug treatment and rehabilitation under the Drug Addiction Treatment Centres Ordinance between July 2017 and March 2018. Ethical approval was obtained from the Institutional Review Boards of the City University of Hong Kong. Written consent was obtained from participants to confirm their willingness to participate. To maintain privacy, the respondents self-administered and completed the questionnaires anonymously. Trained research assistants helped those with poor vision or literacy complete both consent forms and the questionnaire, by reading aloud and explaining their content. 

### 3.2. Participants

Male participants dominated the present study (75.4%, N = 208). The largest age group was 30–39 (51.8%, N = 143). Regarding marital status, 57.3% (N = 157) of participants were single, and 62.3% (N = 167) had no children. Participants had a generally low educational background, and only 3.3.% of them had received post-secondary education. The employment status before incarceration was that around half were employed and half were not (53.6% and 46.4%) (see Table 2).

More than 70% (76.6%, N = 206) of participants first took drugs between 10 and 19 years old, during adolescence. More than 60% (63.5%, N = 167) of them had taken drugs for 1–12 years. In relation to the types of drugs consumed, they mostly took methamphetamine (65.7%, N = 180), cocaine (39.4%, N = 108), and ketamine (33.9%, N = 93). 

Participants had experienced various life issues in the previous five years, while family problems were reported as a major problem by most of the participants (N = 151) (see Table 3). Regarding the impact of the main problems from the perspective of the participants, the top three main ones which the participants perceived as those having most impact were ‘feeling lonely’, ‘relationship problems’, and ‘emotional/mental problems’ (M = 7.0 out of 10). For participants, relationship issues and the corresponding emotions aroused were major problems in their lives that greatly influenced their well-being, and which contributed to the continued use of substances. 

### 3.3. Constructs and Measures

Apart from the demographic background (e.g., gender, age, marital status, number of children, education level, and employment status before incarceration), history and experience of drug use (e.g., first time using drugs, length of taking drugs, types of drugs taken, reasons for taking drugs, etc.) were also included. The questionnaire contained six Likert scales: Meaning of Life Scale (MLS), Reasons for Quitting (RFQ), General Self-Efficacy Scale (GSES), Aspects of Identity Questionnaire (AIQ-IV), Social Isolation Scale (SIS) in the Alienation Scale, and a scale that was created for this study, Soulmate Scale (SS). The present study shortened the length of three scales to ensure the motivation of drug users to complete the questionnaire. To serve specific research purposes and fulfil particular needs of participants, modifications or simplifications on existing scales are common among social sciences research (i.e., [44,45]). Given that about 95% of the participants had been educated to secondary school level or below, undereducated participants perceived a heavy burden for the longer questionnaire, predicting a lower response rate, and more answers lacking a response [46,47]. According to Romppel et al. [48], the proportion between time savings and loss of reliability needed to be considered to ensure the preservation of content coverage. Considering the differences between Western and Eastern contexts, the construction of simplified versions was conducted by four scholars with expertise in Hong Kong drug issues. By listing out all items, the research team considered the used items as the most appropriate measures in the local contexts. Mean and Cronbach’s alpha in the present study is shown. The Cronbach’s alpha of each scale in the present study is similar or even higher than other studies; 0.6 or above was an acceptable value to adopt a scale, and most of the adopted scales demonstrated excellent reliability [49] (see Table 4). To further ensure the validity, Exploratory Factor Analysis (EFA) of each simplified scale was performed, shown in Table 4. The Meaning of Life Scale, which does not simplify, demonstrates the lowest coefficient, but the simplified scales are all above 0.32, representing an acceptable value to capture the dimension of the original scale tapping on [50]. Hence, the validities of these simplified scales are supported, that is, the extracted items can well measure the corresponding variable. Regarding the needs of participants, verified reliability and validity, the scales are justified to be used in the present study. Details of the scales are outlined below (see Table 5). 

Reasons For Quitting Scale. The Reasons for Quitting Scale (RFQ) is a tool used to measure the desire to quit drugs [56]. Twelve items were adopted to conduct multidimensional measures by capturing self-control, health concerns, immediate reinforcement, social influences, and legal motivation for cessation [56]. The RFQ is a five-point scale ranging from 1 for ‘certainly not’ to 5 for ‘certainly’. The mean was 3.93, demonstrating a relatively high motivation to quit. RFQ was also confirmed to have excellent internal consistency with an alpha of 0.907. 

Aspects of Identity Questionnaire-IV (AIQ-IV). AIQ-IV was adopted to measure the Autonomy in Self-determination theory [57]. Given the participants’ background, 35 items would yield a poor response from them. Fourteen statements from three subsections, including self, interpersonal, and others, were extracted with a strong internal consistency (α = 0.911). A five-point scale was used where 1 referred to ‘not important’, and 5 meant ‘very important’. The composite mean was 48.88 out of 70, implying a slight agreement of identity being important to participants.

Social Isolation Scale. Meaningful connections with others enhance the element of relatedness in Self-determination Theory. Based on the Social Isolation Scale of Dwight Dean’s Alienation Scale, five statements were adopted, including ‘Most people today seldom feel lonely’, ‘Real friends are very easy to find’, ‘One can always find friends if he shows himself friendly’, ‘The world in which we live is basically a friendly place’, and ‘People are just naturally friendly and helpful’ [58]. These authors commented that the high discriminative power of the items and the T-values of subscales allowed the independent use of each subscale and item, which may even ‘yield better and more meaningful result[s]’ [58] (p. 89). Statements could be ranked from 1 to 5 (1 = strongly disagree; 5 = strongly agree). Each score of the answers was summed to represent the level of social isolation. The mean score was 14.18 out of 25. An acceptable value of internal consistency was tested to be 0.634.

Soulmate Scale. Psychological connections with drugs, namely soulmate experiences, were recently confirmed as gaining comfort, a sense of security and satisfaction [12]. The scale covered 12 items for three factors, including (1) psychological release and shelter, (2) staunch and supportive friendship, and (3) spiritual solace and companionship. Statements could be ranked from 1 to 5 (1 = strongly disagree; 5 = strongly agree). In the present study, the composite mean scored 2.74. An excellent internal consistency, 0.931, was obtained. 

Meaning of Life Scale. This scale contains two sub-scales (presence and searching for meaning), but Demirbaş-Çelik and Keklik [59] found that the search for meaning does not correlate with SDT. Therefore, the five-item presence meaning scale was selected to shorten the length of the questionnaire, and provide an overview of psychological health, including the inverse relationship with certain psychological disorders such as depression and anxiety [54]. The Cronbach’s alpha was calculated to be 0.786, suggesting high internal consistency. The composite mean was 21.54 out of 30, suggesting participants had found some meaning in their lives.

General Self-Efficacy Scale. Regarding the framework, competence is similar to self-efficacy [18]. The Chinese version of the General Self-Efficacy Scale was demonstrated to have an excellent internal consistency, and its reliability was not affected by eliminating items [60]. Five statements were extracted to reduce the respondents’ burden (‘I can always manage to solve difficult problems if I try hard enough’, ‘It is easy for me to stick to my aims and accomplish my goals’, ‘I am confident that I could deal efficiently with unexpected events’, ‘I can remain calm when facing difficulties because I can rely on my coping abilities’, and ‘If I am in trouble, I can usually think of a solution’). Respondents could rank each statement between 1 to 4 where 1 refers to ‘not at all true’ and 4 means ‘exactly true’. A highly reliable Cronbach’s alpha (α = 0.854) demonstrated the internal consistency among items. The composite mean was 13.33 out of 25, indicating fairly high confidence in participants’ capability. 

### 3.4. Analyses

To begin with, inter-variable correlations were performed to investigate the significance of the relationships between the variables. Next, multiple regression analysis was performed to see which variables, including a sense of identity, soulmate experience, self-efficacy, the meaning of life, and sense of social isolation, were predictive of drug-taking. Last but not least, mediation analysis was performed to see how soulmate experience, sense of social isolation, and meaning of life were related to drug-taking. The above analyses were performed using SPSS 26.

## 4. Results

Correlation analysis was performed to investigate how participants’ sense of identity, self-efficacy, meaning of life, and social isolation were related to their soulmate experience in drug-taking and reason for quitting drugs. As shown in Table 6, a sense of identity (*r* = −0.150 *), social isolation (*r* = −0.211 **), and meaning of life (*r* = −0.369 **) were significantly negatively correlated to soulmate experience. Meanwhile, sense of identity (*r* = 0.366 **), self-efficacy (*r* = 0.212 **), and meaning of life (*r* = 0.317 **) were significantly positively correlated to quitting drugs, while soulmate experience was significantly negatively correlated to quitting drugs (*r* = −0.389 **). These results show that when one has not developed one’s identity and self-efficacy, found a purpose in life, and experienced a sense of social isolation, one will have a greater dependence on drugs as a soulmate, and vice versa. These results support Lo and colleagues’ [12] literature about the relationship between a sense of loneliness and dependence on drugs as a soulmate. 

In order to see the relative importance of each variable in affecting drug-taking, a multiple regression analysis was performed to see how soulmate experience, meaning of life, aspect of identity, sense of efficacy, and social isolation predicted quitting drugs (see Table 7). Results showed that soulmate experience (β = −0.392 ****) and social isolation (β = −0.202 ***) were significantly negatively predictive of quitting drugs, while aspect of identity (β = 0.277 ****) was significantly positively predictive of quitting drugs. Meaning of life and general self-efficacy showed no significance in predicting quitting drugs. This shows that ‘autonomy’ and ‘relatedness’ are particularly relevant in affecting one’s growth and development [16] (p. 68). Feeling autonomous in making life decisions and achieving certain goals, as well as the quality of social connections affects one’s tendency to engage in drugs [15]. Lacking a sense of *autonomy* and *relatedness* will instigate one to get attached to drugs in order to seek psychological fulfilment and comfort, and this resembles the soulmate experience [12,15]. Additionally, among all the variables, soulmate experience was the most predictive of quitting drugs. This shows the significance of soulmate experience in affecting one’s tendency to take drugs. As stated by Lo and colleagues [12], ‘the motivation for using substances will further be boosted’ when ‘substance users can obtain a sense of belonging or “being loved” through the use of substances’ to ‘relieve pains or fulfil fantasies’ (p. 2). The ‘emotional tie’ built with drugs [12] (p. 2) encourages drug users to engage in drugs, thus lowering their tendency to quit.

The importance of soulmate experience was further investigated through mediation analyses, which were performed to see if soulmate experience significantly mediated the effect of social isolation, meaning of life, sense of identity, and self-efficacy on quitting drugs. Mediation paths of ‘social isolation/meaning of life/aspect of identity/general self-efficacy → soulmate experience → quitting drugs’ were established. The mediation properties of outcome expectancies of the mediation paths were investigated. 

Results presented in Figure 2, Figure 3, Figure 4 and Figure 5 show that while the mediation path ‘general self-efficacy → soulmate experience → quitting drugs’ was not significant (Figure 5), all the other three proposed mediation paths were significant. For the mediation path, ‘general self-efficacy → soulmate experience → quitting drugs’, self-efficacy did not significantly anticipate soulmate experience (*p* > 0.05), but soulmate experience negatively anticipated quitting drugs (β = −0.319 ***). Self-efficacy positively anticipated quitting drugs (β = 0.047 ***); with soulmate experience as the mediator, the effect of self-efficacy on quitting drugs was diminished (β = 0.042 ***). This suggests that having a high level of self-efficacy—which indicates a sense of competence and capability to attain the goals [16,17,18]—does not result in one’s attachment to drugs as a way to seek comfort, which in turn affects quitting drugs; meanwhile, it facilitates one to quit drugs. 

For the mediation path, ‘aspects of identity → soulmate experience → quitting drugs’ (Figure 2), aspects of identity negatively anticipated soulmate experience (*β* = −0.014 *), and soulmate experience negatively anticipated quitting drugs (*β* = −0.289 ***). Aspects of identity positively anticipated quitting drugs (*β* = 0.029 ***); with soulmate experience as the mediator, the effect of aspects of identity on quitting drugs was diminished (*β* = 0.025 ***). 

For the mediation path, ‘social isolation → soulmate experience → quitting drugs’ (Figure 3), social isolation negatively expected soulmate experience (*β* = −0.054 ***), and soulmate experience negatively expected quitting drugs (*β* = −0.376 ***). Social isolation negatively expected quitting drugs (*β* = −0.037 **); with soulmate experience as the mediator, the effect of aspects of identity on quitting drugs was enlarged (*β* = −0.057 ***). 

For the mediation path, ‘meaning of life → soulmate experience → quitting drugs’ (Figure 4), meaning of life negatively expected soulmate experience (*β* = −0.071 ***), and soulmate experience negatively expected quitting drugs (*β*= −0.266 ***). Meaning of life positively expected quitting drugs (*β* = 0.052 ***); with soulmate experience as the mediator, the effect of meaning of life on quitting drugs was diminished (*β* = 0.033 ***). 

The significance of the mediation paths ‘aspects of identity → soulmate experience → quitting drugs’ (Figure 2), ‘social isolation → soulmate experience → quitting drugs’ (Figure 3), and ‘meaning of life → soulmate experience → quitting drugs’ (Figure 4) indicated that hypotheses 1, 2, and 3 were supported. Soulmate experience significantly mediated the relationship of social isolation and meaning of life, apart from aspects of identity, with soulmate experience. In addition, the above results show that the mediating effects of the mediation paths ‘social isolation → soulmate experience → quitting drugs’ (Figure 3) and ‘meaning of life → soulmate experience → quitting drugs’ (Figure 4) were greater than that of ‘aspects of identity → soulmate experience → quitting drugs’ (Figure 2). The path ‘social isolation → soulmate experience → quitting drugs’, in which the direct effect (c’) was larger than the total effect (c) even indicated a suppression effect [61]. These results suggest that the feeling of loneliness is a strong drive for taking drugs to seek comfort [12]. While drugs, for users who experience loneliness, are regarded as soulmates offering comfort for them [12], if such a sense of being cared for, understood, and loved could be achieved from elsewhere (e.g., from relationships with significant others), such a sense of psychological fulfilment could play a part in helping individuals find their purpose of life, thus encouraging them to quit drugs. 

## 5. Discussion

Results of correlation and regression analysis showed that among several variables (i.e., sense of identity, meaning of life, self-efficacy, social isolation, and soulmate experience), soulmate experience was the most related to drug-taking behaviour. Having soulmate experience lowered the tendency to quit drugs and encouraged continuous drug-taking. This supports Lo and colleagues’ [12] notions about the connections between substances and substance users. Experiencing a lack of connectedness with significant others, and the psychological need for belonging has not been fulfilled [19,20]. Hence, drugs become an alternative, the soulmate for users to get attached to, achieve psychological compensation, and alleviate the sense of loneliness and negative emotions [12]. Moreover, while a sense of identity and self-efficacy, and meaning of life (e.g., having one’s own life goals and sense of self-worth, having a sense of competence and the autonomy to quit drugs because of a recognition of the negative effects of drug-taking) were related to quitting drugs [15], in this study, these three factors were found to be less significantly related to quitting drugs. This shows that the psychological states of drug users are a significant issue when investigating their tendencies to quit drugs, and when designing corresponding drug treatment plans for them. 

In addition, mediation analysis showed that soulmate experience significantly mediated the effect of meaning of life and social isolation, besides a sense of identity, on quitting drugs. This reflects that the spiritual and emotional bonding with others help dispel the loneliness, and bring about meaning of life for individuals. A soulmate is someone with whom one is willing to have a spiritual bond, and from which one can achieve unconditional positive regard, warmth, comfort, relief, and a sense of security in the alleviation of negative emotions [12]. Having connections with and being valued by other people help fulfil the needs for belonging and to achieve a sense of well-being and sense of meaning of life (e.g., [62]). If drug users cannot find such meaningful connections in life, they will turn to drugs to seek a sense of warmth, support, and spiritual companionship as an alternative [12,15]. These results provide valuable implications for service regarding the importance of helping drug users nurture quality and supportive relationships with significant others in order to facilitate their rehabilitation and reduce the chances of relapse. 

## 6. Conclusions

Using the Soulmate Scale [12], this study helps enrich the knowledge base regarding the investigation of the psychological experience of drug users, and the underlying psychological factors affecting their drug-taking behaviour, as well as their tendency to quit drugs. Results of this study support previous findings of studies that ‘relatedness is an essential element that determines drug users’ choice to take or to quit drugs’ [15] (p. 14). Owing to the significance of spiritual sustenance, psychological fulfilment, and the sense of being supported and positively regarded in relationships [12,15], it is important to deal with the psychological states of the drug users and the drug rehabilitators, so as to facilitate their recovery and their ability to lead a healthy, adaptive life. Social services can focus more on dealing with their sense of loneliness, such as offering more individual counselling to alleviate their individual ‘emotional loneliness’, as well as providing groupwork to deal with their ‘social loneliness’ [12] (p. 12). Individual counselling can adopt a person-centred approach to show more care and understanding of the underlying reasons for drug-taking [42]. Groupwork helps drug users build connections with adaptive peers to replace their drug-taking networks [12]. Last but not least, family therapy/counselling can be constantly provided for drug users and rehabilitators. Acceptance by the family is ‘a powerful factor’ which enhances the drug users’ and rehabilitators’ ability ‘to continue their treatment and stay away from drugs’ [63]. In this vein, family therapy/counselling can focus on nurturing support, emotional bonding, a sense of cohesiveness and happiness, and adaptive communication patterns, so as to increase the success of drug rehabilitation and reduce the chances of relapse [63].

## 7. Limitations

One limitation comes from the sample of study. The participants were solely recruited from Drug Addiction Treatment Centres, raising the issue of generalisability of the results to the drug-taking population from other localities. In future studies, other sources can be exhausted to recruit drug-taking participants. Next, the fairly low effect size demonstrated in the multiple regression analysis suggests that there might be other significant factors affecting one’s tendency to take drugs or quit drugs which were not included in the model. More studies can be performed in the future to investigate this topic. Last but not least, this study is a cross-sectional one, meaning that the participants’ psychological experience of drug-taking was merely captured at a particular point of time. As drug-taking is a “chronic relapsing disorder” in which the drug takers normally experience recurring affective states and stimuli from the environment [64] (p. 893), in the future, a longitudinal study can be conducted to fully grasp the drug takers’ psychological experiences and their effects on drug-taking/quitting drugs. 

## Figures and Tables

**Figure 1 ijerph-18-12730-f001:**
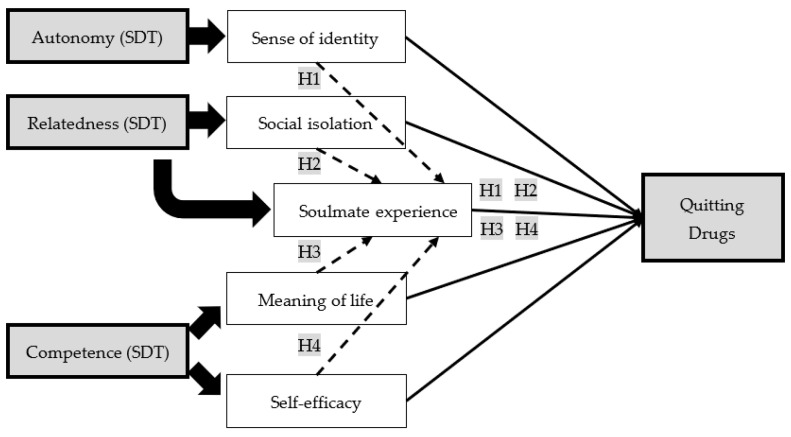
Theoretical framework of this study.

**Figure 2 ijerph-18-12730-f002:**
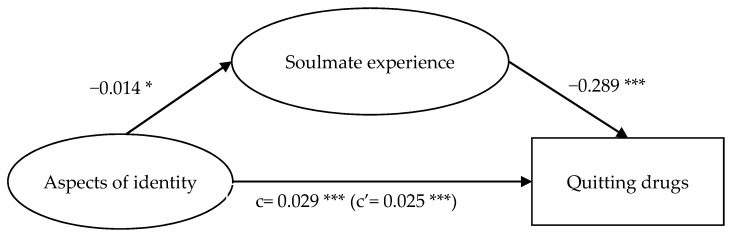
Mediation model of aspects of identity, soulmate experience, and quitting drugs. Note: * *p* < 0.05, *** *p* < 0.001.

**Figure 3 ijerph-18-12730-f003:**
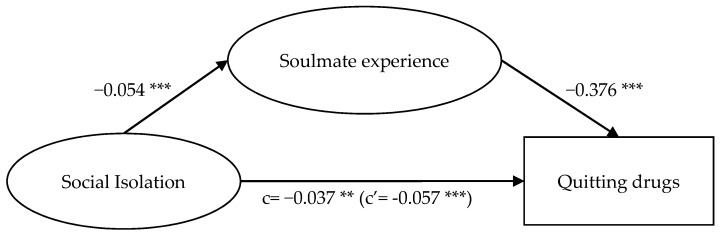
Mediation model of social isolation, soulmate experience, and quitting drugs. Note: ** *p* < 0.01. *** *p* < 0.001.

**Figure 4 ijerph-18-12730-f004:**
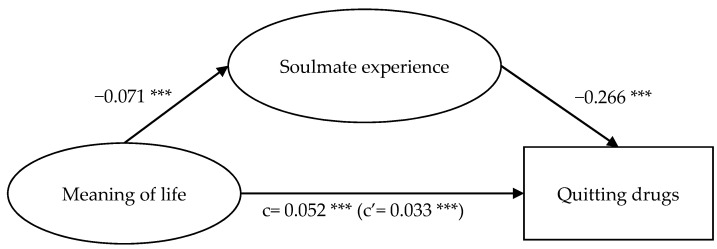
Mediation model of meaning of life, soulmate experience, and quitting drugs. Note: *** *p* < 0.001.

**Figure 5 ijerph-18-12730-f005:**
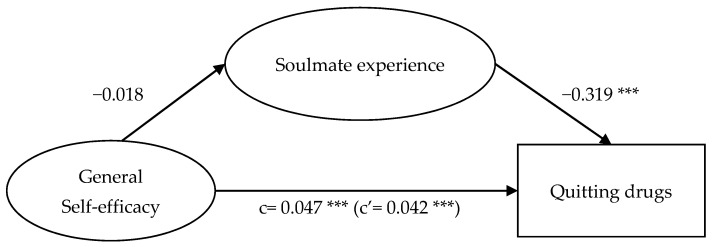
Mediation model of general self-efficacy, soulmate experience, and quitting drugs. Note: *** *p* < 0.001.

**Table 1 ijerph-18-12730-t001:** Numbers of participants from each correctional institution.

Institution	Participants’ Status	No. of Participant	Total Number of Questionnaires
CI-1	Inmate	95	190
	Recallee	75	
	Supervisee	20	
CI-2	Inmate	34	65
	Recallee	20	
	Supervisee	11	
CI-3	Inmate	13	18
	Recallee	5	
CI-4	Inmate	3	3
Total			276

**Table 2 ijerph-18-12730-t002:** Participants’ demographic data.

Variable	No. of Participants (N)	%
Gender (N = 276)		
Men	208	75.4
Women	68	24.6
Age (N = 276)		
16–20	10	3.6
20–29	123	44.6
30–39	143	51.8
Marital status (N = 274)		
Single	157	57.3
Married	56	20.4
Living together	28	10.2
Divorced	24	8.8
Separated	4	1.5
Widow/widower	3	1.1
Remarried	1	0.4
Others	1	0.4
No. of children (N = 268)		
No children	167	62.3
1 child	56	20.9
2 children	29	10.8
3 or more children	16	6
Highest education attained (N = 274)		
Junior high (F.1–F.3)	167	60.9
Senior high (F.4–F.6)	74	27.0
Elementary or below	19	6.9
Associate degree	5	1.8
Bachelor degree	4	1.5
Matriculation (F.6–F.7)	2	0.7
Master degree or higher	2	0.7
Others	1	0.4
Employment status before incerceration (N = 274)		
Unemployed	127	46.4
Employed	147	53.6
Age at first drug-taking (N = 269)		
10–19 years old	206	76.6
20–29 years old	54	20.1
30–39 years old	9	3.3
Length of taking drugs (N = 263)		
Fewer than 1 year	5	1.9
Between 1 and 4 years	54	20.5
Between 5 and 8 years	56	21.3
Between 9 and 12 years	57	21.7
Between 13 and 16 years	44	16.7
Between 17 and 20 years	31	11.8
More than 20 years	16	6.1
Types of drugs taken (could choose more than one option) (N = 274)		
Methamphetamine (Ice)	180	65.7
Cocaine (Cola)	108	39.4
Ketamine (Kai)	93	33.9
Marijuana	41	15.0
Nimetazepam	39	14.2
Heroin	36	13.1
Zopiclone/Triazolam	28	10.2
MDMA	25	9.1
Cough syrup	23	8.4
Others	7	2.6

**Table 3 ijerph-18-12730-t003:** Main problems in past 5 years and the impact (could choose more than one option).

Main Problems in Past 5 Years	No. of Participants (*N*)	M
Feeling lonely	78	7.0
Relationship problems	121	7.0
Emotional/mental problems	101	7.0
Family problems	151	6.9
Work/occupational problems	85	6.5
Financial problems	124	6.4
Gambling problems	56	6.2
Residential problems	69	6.0
Alcoholism problems	39	5.5
Triad problems	52	5.5
Academic problems	33	4.2
Others	26	5.6

Note: The impact of each of the main problems experienced was rated on a 10-point Likert scale (0 = absolutely no impact; 10 = absolutely impactful).

**Table 4 ijerph-18-12730-t004:** The mean and Cronbach’s alpha of the six scales.

Scale	No. of Items	Mean	Cronbach’s Alpha	Cronbach’s Alpha in Other Studies
Reason For Quitting ^1^	12	3.93	0.907	0.74–0.82 [51]
Aspect of Identity Questionnaire-IV ^2^	14	48.88	0.911	0.88 [52]
Social Isolation ^3^	5	14.18	0.634	0.70 [53]
Soulmate Scale ^3^	12	2.74	0.931	0.933 [12]
Meaning of Life Scale ^5^	5	21.54	0.786	0.86 [54]
General Self-Efficacy Scale ^4^	5	13.33	0.854	0.69 [55]

Note: ^1^ 1–5 points (1 = certainly not; 5 = certainly); ^2^ 1–5 points (1 = not important; 5 = very important); ^3^ 1–5 points (1 = strongly disagree; 5 = strongly agree); ^4^ 1–4 points (1 = not at all true; 4 = exactly true); ^5^ 1–6 points (1 = absolutely untrue; 6 = absolutely true).

**Table 5 ijerph-18-12730-t005:** The coefficients of EFA of the six scales.

Scale	Factor Loading
Reasons For Quitting	0.408–0.773
Aspects of Identity	0.333–0.728
Social Isolation	0.392–0.740
Soulmate	0.674–0.797
Meaning of Life	0.264–0.706
General Efficacy	0.605–0.783

**Table 6 ijerph-18-12730-t006:** Inter-variable correlations of measurement scales.

Measure	Soulmate	Quitting Drugs
Soulmate	-	−0.389 **
Aspect of identity	−0.150 *	0.366 **
General self-efficacy	−0.067	0.212 **
Meaning of life	−0.369 **	0.317 **
Social isolation	−0.211 **	−0.169 **

Note: * *p* < 0.05. ** *p* < 0.01.

**Table 7 ijerph-18-12730-t007:** Multiple regression estimates for quitting drugs.

	Unstandardised Coefficients		Standardised Coefficients	R^2^	Adjusted-R^2^
	B	Std. Err	Beta		
Soulmate experience	−0.332 ****	0.049	−0.392 ****	0.282	0.269
Meaning of life	0.000	0.011	−0.002		
Aspect of identity	0.022 ****	0.005	0.277 ****		
General-efficacy	−0.004	0.014	−0.019		
Social isolation	−0.044 ***	0.013	−0.202 ***		

Note: *** *p* < 0.001, **** *p* = 0.0000.

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
