# Peer review of "An Inquiry into the Relationship between Drug Users’ Psychological Situations and Their Drug-Taking Behaviour"

_ijerph, 2021, doi:10.3390/ijerph182312730_

Round 1
Reviewer 1 Report
This is an interesting study and, generally, a well written manuscript. However, there are a number of issues that need attention, as follows:
- Variable names and meanings
- Throughout the Introduction 'autonomy' is referred to as a variable, but then in Figure 1 this variable is labelled 'sense of identity', and the Methodology section states that 14 items from the AIQ-IV were used to measure 'autonomy', but also says that the mean score implies 'a slight agreement of identity being important to participants', followed by 'aspect of identity' being used throughout results. This all gets very confusing - if the variable you measured is 'autonomy', then refer to the variable as 'autonomy' throughout the manuscript.
- Similar issues are evident for the variable measured via the RFQ - which is referred to, for example, as reflecting 'the desire to quit substances', 'Reasons for Quitting', 'duration of drug use (line 379)', 'quitting drugs', 'drug taking' and 'one's drug abstinence'. All of these things have different meanings, hence it is essential that a consistent variable name is used and that it is clear to the reader what this variable actually measures. From the measure used, presumably you should be referring to 'reasons for ceasing drug use' or similar.
- You have written 'locus of control' rather than 'self-efficacy' in line 370
- Regarding the shortened measures
- using just the presence scale from the Meaning of Life Scale is fine
- more information is required regarding how the 14 items were chosen from the AIQ-IV and which items they are. In particular, some justification is required for why these specific items and number of items (i.e., why not more/fewer or different items) were selected
- similarly, while you have included mention of the five items selected from the General Self-efficacy scale, information how/why these specific item were selected
- Results
- I'm unsure as to why you presented both bivariate correlations and linear regressions for 'Reasons for quitting'/'Quitting drugs' as the results are basically the same - would make more sense to follow the correlations with a multiple regression analysis including all 5 variables concurrently.
- many of the pathway weights in the mediation models look quite low in relation to the significance assigned to them - please check these are accurate
- similarly, interpretations of findings should place some weight on effect sizes, not just statistical significance levels - for example, the correlation between 'aspect of identity' and 'soulmate' is -.15, so only 2% shared variance; while significant at .05 level, this is a weak effect
- I think the text in the Results section could be shortened by just referring to the Figures rather than also including the mediation pathways - it's not very reader-friendly in it's current format
- Discussion
- you have not made any mention of the limitations affecting your study - in particular, you have not interpreted your results taking effect sizes in to consideration (as noted above), nor is there seemingly any consideration for the limitations associated with cross-sectional data
Reviewer 2 Report
Drug abuse or substance abuse refers to the use of certain chemicals for the purpose of creating pleasurable effects on the brain. There are over 190 million drug users around the world and the problem has been increasing at alarming rates, especially among young adults under the age of 30. Apart from the long term damage to the body drug abuse causes, drug addicts who use needles are also at risk of contracting HIV and hepatitis B and C infection.
Therefore, the manuscript submitted for review presents a current and clinically relevant problem. The manuscript is correctly structured in accordance with the requirements of scientific work. The introduction is a comprehensive and interesting introduction to the topic, the methodology is described in an accurate and meritorious way. The results are correctly and accurately described. However, in my opinion literature review is not necessary in the manuscript and its removal will make it clearer and more consistent. Similarly, the Authors should considerably shorten conclusions, to a maximum of 2-3 sentences
Author Response
Thanks for your comments.
The literature review provided a foundation for developing the hypotheses. Also, the conclusion had outlined the significant implications of the present study.
Reviewer 3 Report
Interesting text.
It would be more interesting if the conclusions were followed by a point about practical implications in rehabilitation programs and in primary and secondary prevention programs.
Author Response
Thanks for your comments.
Recommendations for rehabilitation programs and counselling services were discussed in conclusion.